environmental science

sustainable consumption, value-sensitive design, attitude-behaviour-gap

**Author for correspondence:**
Johannes Klinglmayr
e-mail: johannes.klinglmayr@lcm.at

# How value-sensitive design can empower sustainable consumption

Thomas Asikis[1], Johannes Klinglmayr[2], Dirk Helbing[1] and Evangelos Pournaras[3]

[1]Professorship of Computational Social Science, ETH Zurich, Zurich, Switzerland
[2]Linz Center of Mechatronics GmbH, Linz, Austria
[3]School of Computing, University of Leeds, Leeds, UK

 TA, 0000-0003-0163-4622; JK, 0000-0002-1665-7803;
DH, 0000-0002-9898-0101; EP, 0000-0003-3900-2057

In a so-called overpopulated world, sustainable consumption is of existential importance. However, the expanding spectrum of product choices and their production complexity challenge consumers to make informed and value-sensitive decisions. Recent approaches based on (personalized) psychological manipulation are often intransparent, potentially privacy-invasive and inconsistent with (informational) self-determination. By contrast, responsible consumption based on informed choices currently requires reasoning to an extent that tends to overwhelm human cognitive capacity. As a result, a collective shift towards sustainable consumption remains a grand challenge. Here, we demonstrate a novel personal shopping assistant implemented as a smart phone app that supports a value-sensitive design and leverages sustainability awareness, using experts' knowledge and 'wisdom of the crowd' for transparent product information and explainable product ratings. Real-world field experiments in two supermarkets confirm higher sustainability awareness and a bottom-up behavioural shift towards more sustainable consumption. These results encourage novel business models for retailers and producers, ethically aligned with consumer preferences and with higher sustainability.

## 1. Introduction

Creating more sustainable consumption patterns turns out to be imperative for mitigating climate change and supporting a more viable future of our society [1,2]. Food systems, in particular, play a key role, influencing 12 out of the 17 sustainable development goals of the United Nations [3]. Food supply is already leading total greenhouse gas emissions [4,5], while food demand is expected to grow by 50% along with an increase of the global population from 7 to 9.8 billion people by 2050 [6]. Thus,

**Figure 1.** An outline and comparison of the proposed value-sensitive, preference-based decision-support process. (*a*) A consumer's localization at Retailer A. The red dots on the map denote the product categories in close proximity. (*b*) Presenting the product categories in close proximity to the consumer. (*c*) Product ratings close to 10 denote better matching of the product to the consumer's sustainability preferences. (*d*) Comparison of the proposed value-oriented, preference-based design approach with mainstream decision-support shopping assistants.

introducing policies that promote more sustainable consumption patterns is critical. Yet, it has proven to be insufficient to overcome the 'attitude-behaviour gap' [7]. For instance, choosing grocery products according to a broad spectrum of (often opposing [8–10]) sustainability criteria requires the processing of an overwhelming amount of information and, as a result, the price often remains the dominant choice criterion. Such information is often summarized and its processing is automated, i.e. by product labels or smart phone shopping assistants that scan product barcodes [11–13]. However, ensuring a value-sensitive design in terms of accountability, transparency, privacy, self-determination and an overall practicality by a seamless integration into the shopping process remains a grand challenge. However, designing digital shopping assistants, e.g. smart phone apps, for such values turns out to be a requirement for consumer trust (see IEEE Global Initiative on Ethics of Autonomous and Intelligent Systems [14]). Given the broad scope and complexity of the subject of sustainability and the co-determination of consumer decisions by price [15], product choices based on sustainability criteria can be particularly prone/vulnerable to manipulative nudging when personalized decision-support systems use personal data.

In this contribution, we show how value-sensitive design can empower a bottom-up shift to more sustainable consumption. To make this possible, we built a novel and general-purpose shopping assistant [16] and tested it at two real-world supermarkets in Estonia (Retailer A [17]) and Austria (Retailer B [18]). The shopping assistant was implemented as an Android smart phone application (app) for decision-support: it rates products (as well as product categories) that are in front of the consumer in a shop according to *personal sustainability criteria* (preferences) and in a privacy-preserving way. These preferences cover the categories of *environment*, *social*, *health* and *(product) quality* so that consumers can improve their health while supporting socially favourable production conditions and making choices with better environmental impact. Products that better match consumers' preferences are rated close to 10 and those that oppose the preferences receive values close to 0. The consumer can interact with the app to change sustainability preferences using a continuous Likert-scale slider in the range [0, 10]. Ratings are explainable to consumers by making transparent why certain products receive higher ratings than others.

The value-sensitive design of this privacy-preserving digital shopping assistant is distinguished from other mainstream approaches (figure 1*d*) by integrating the following values in decision-support: (i) adding environmental, social, health and quality criteria in consumers' decisions, (ii) limiting manipulative nudging, (iii) self-determination in consumers' preference choices, (iv) privacy preservation, (v) explainability, and (vi) practicality and compatibility with the existing shopping process.

These values are realized by a design process that tackles the challenges to provide (i) *transparent and explainable personalized product ratings*, (ii) *a seamless integration into the consumers' shopping process*, and

(iii) *scalable, high-quality product information on sustainability*. The scope and combination of inter-disciplinary methods required for this is quite exceptional, covering a new value-sensitive approach, its implementation within a sophisticated system and the testing of this approach in real-world settings using rigorous social science methods. This sets this work apart from most other state-of-the-art studies (see electronic supplementary material, table S.6).

*Transparent and explainable personalized product rating.* The first challenge of value-sensitive design is addressed by designing a novel content-based recommender algorithm for the personalized rating of products that is decentralized and privacy-preserving. In contrast to user-based collaborative filtering algorithms that rely on the collection of sensitive historical consumer data to compare purchase profiles [19], the novelty of our algorithm is that personalization is taking place on the smart phone and as a result, the sustainability preferences of a consumer remain localized on the consumer's device, i.e. private. This is possible via a scalable, distributed communication protocol that handles a consumer's request for public summarized product information on sustainability (sustainability indices). When this information is localized on the smart phone, efficient calculations turn the sustainability index of products into personalized product ratings. The implications of our value-oriented and preference-based design approach are significant: without sharing sensitive personal data with third parties, manipulative nudging that serves corporate interests and may oppose personal preferences or sustainable consumption patterns is limited [20]. As a result, consumers can trust that the rating of the products is a result of their own intrinsic values expressed via their sustainability preferences. Moreover, the calculation of a product rating is accountable to the consumer, who can visually explore which product information and preferences are mainly responsible for the overall rating (see electronic supplementary material, figure S.13). The explainability is also localized on the smart phone so that no third parties can manipulate the perception of a consumer on why certain products receive higher or lower ratings.

*Seamless integration into the consumer shopping process.* For the second challenge, a seamless integration in the shopping process is crucial for the practicality and adoption of the solution in retailer shops. Barcode scanning makes the comparison of different products cumbersome [21]. Instead, in our system, consumers automatically view all nearby products of a category, e.g. all different pasta products, on their smart phone and can therefore make efficiently an augmented comparison while moving along the shelves (see figure 1; electronic supplementary material, figure S.12). This novel augmented reality experience is made possible via a low-power bluetooth beacon localization technology that has been deployed and extensively tested on retailers' sites [22]. Consumer localization in the shop does not require a centralized collection of GPS traces which tends to be unreliable, privacy-intrusive and not suitable for indoor environments [23].

*Scale and quality of product information on sustainability.* This third challenge is addressed by designing and populating an open knowledge-base. It is based on a new sustainability ontology with which a transparent and accountable reasoning for the personalized rating of products is performed. Ontologies have been influential and have often accelerated scientific progress in biology and beyond [24–26]. They systematically dissect and structure complex concepts to reason and develop a shared understanding that motivates their use in this work. The designed knowledge-base consists of information from retailers, online data sources with domain and crowd-sourced knowledge, domain experts, who supported this project during its lifetime, as well as the wisdom of crowds [27,28], for example, by crowd-sourcing the analysis of Wikipedia data (see electronic supplementary material, table S.7). The knowledge-base is actually a set of 795 associations between 15 472 products and 25 consumer preferences belonging to the four sustainability categories of environment, health, social and quality. See electronic supplementary material, table S.1–S.4 for a complete list of all preferences and how they are obtained. Formally, the association between a product keyword (e.g. meat) and a preference keyword (e.g. vegetarian) is quantified in the range $[-1, 1]$, with $-1$ representing opposition (meat does not fit vegetarian diet) and 1 representing support (lettuce fits vegetarian diet). In-between values represent, for instance, the relative positive effect of different vitamins or the relative negative effect of different additives/preservatives in several health indicators. Therefore, the design of the knowledge-base is the selection of such keywords (tags) according to formal semantics as well as the reasoning about their in-between association score (196 product and 61 preference tags as shown in electronic supplementary material, figure S.4). This semantic information is used to calculate a non-personalized *sustainability index* of a product for certain sustainability preferences by aggregating the association scores among the assigned product and preference tags. Moreover, rebound effects that model the incompatibility of sustainable development goals [8–10] at the level of consumer choices can be measured at the design phase by analysing the semantic links that interconnect products, preferences and their tags.

*Field tests.* We conducted novel and ambitious field tests at two supermarkets from May to November 2018. These go beyond earlier survey questionnaires [29–31], laboratory experiments [32] or gamification [33]. We implemented the actual value-sensitive and preference-based decision-support system to empower consumers to shift their shopping choices to sustainable consumption. As a result of the actual system implementation, a more realistic assessment of the shopping shift can be made. A total number of 323 (Retailer A) and 69 (Retailer B) participants with a loyalty card were recruited by the retailers and project staff members on site by offering coupon discounts of up to 100 euros. The loyalty card provides access to baseline historical purchases allowing us to compare consumers using the app (treatment condition) with similar consumers not using the app (no treatment). The rewards are designed to incentivize a regular shopping behaviour, limit dropouts and biases. Consumers using the app fill in an entry survey, choose their (baseline) preferences that can be changed later and start shopping with the app. For the purpose of this study, the app usage and purchases are traced but remain anonymized. After the end of the test period consumers collect their coupons by answering an exit survey.

*Outline of analysis.* The data collected during the field tests allows us to analyse whether consumers shift their shopping choices to other products that better respect their own values, self-determined via their sustainability preferences. It turns out that consumers are usually ready to pay more for higher quality products, which are healthier and more sustainable [15]. We dissect the complex link of sustainability preferences with price by analysing whether such a link originates from consumer preferences or from biases in the knowledge-base we developed. We further identify the profiles of consumers' preferences and illustrate to what degree they are adjusted to meet their expectations of how products should be rated.

# 2. Results

We first demonstrate the shift of consumer behaviour towards more sustainable consumption, meaning products that are rated higher than 5. The sustainable shopping behaviour is measured using the weekly expenditures made for such highly ranked products, normalized to [0, 1], using the observed maximum value. The expenditure values are deflated using the harmonized index of consumer prices [34]. The consumers that did not use the app (control group) were matched with consumers that did use the app. The following matching criteria (behaviour covariates) were applied: (i) the deflated monthly total budget spent, (ii) the distribution of deflated monthly budget spent per product category, and (iii) the sustainability index of the average purchased products per month. Matching reduces the control group from 3438 to 532 for Retailer A and from 1843 to 59 for Retailer B. The total consumers who participated in the field tests are filtered out to 148 (Retailer A) and 30 (Retailer B) consumers (treatment) by keeping the consumers who have seen the rating of the products before they have purchased it. The weekly expenditures of the consumers using the app are predicted based on 'historic' data in order to estimate the shopping behaviour of the treatment group in case they did not use the shopping app. This is done via causal impact analysis [35] using spike-and-slab causal inference via structured Bayesian times series [36]. The predicted values are compared to the actual weekly expenditures made when using the app (see figure 2, where the prediction horizon covers the entire field test period).

Our results confirm a statistically significant increase of expenditures for highly rated products: (i) the absolute effect is about 20% for each retailer, and (ii) the relative effect is 36.7% for Retailer A and 41% for Retailer B. The cumulative effect is depicted in order to show how this shift towards expenditures for more sustainable products unfolds over the period of the field test. Electronic supplementary material, table S.10 summarizes the results of the causal impact analysis.

A few additional observations can strengthen the conclusions of figure 2: 68% and 75% of the total purchased products in Retailer A and Retailer B, respectively, are highly rated greater than 5 products that are seen in the app. Novel purchases of (i) products purchased for the first time and (ii) products whose rating is seen in the app also increase, namely, by 22% for Retailer A and 16% for Retailer B.

The survey results of electronic supplementary material, figure S.5, S.6 and S.7 furthermore confirm the following: (i) high product ratings reflect consumers' preferences, (ii) new products are discovered via the products rating, (iii) the rating of products appears justified to consumers, and (iv) consumers are more aware of sustainability aspects after using the app.

Note that the two retailer shops have different weekly expenditure patterns. Retailer A exhibits stronger seasonality effects than Retailer B, which considerably relies on customers with a student

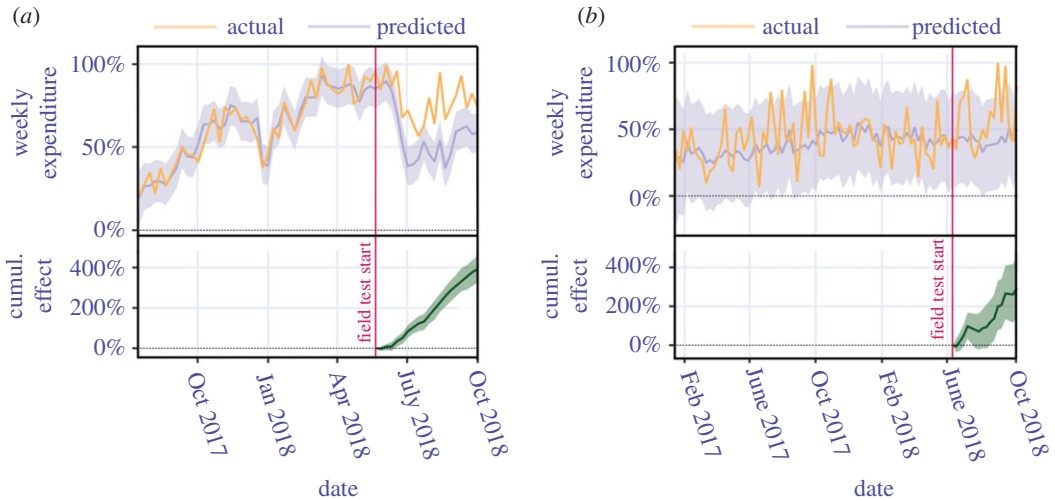

**Figure 2.** Shift to more sustainable consumption during the field test period. Actual versus predicted normalized weekly expenditures made for highly rated products greater than (5). (*a*) Retailer A and (*b*) Retailer B. We find a high statistical significance of $p < 0.001$ for a significance level of $\alpha = 0.01$.

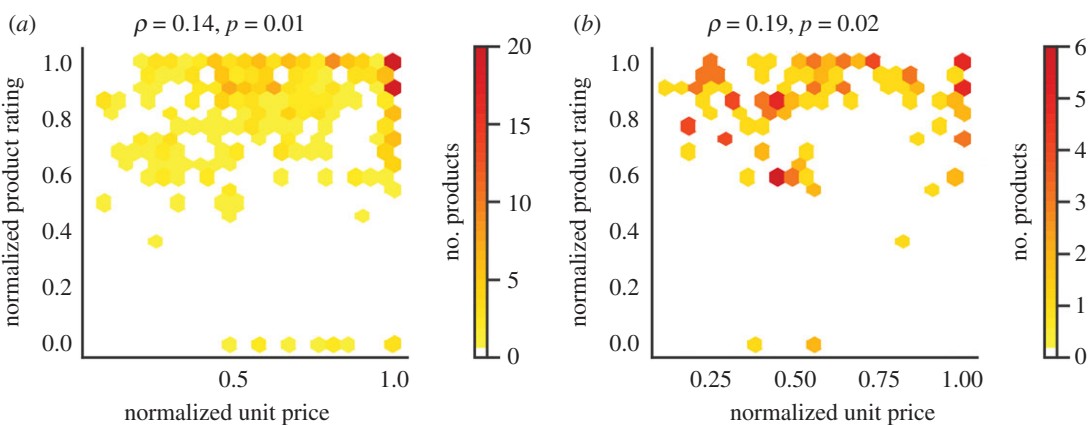

**Figure 3.** Highly rated products correlate to some extent with increased prices. (*a*) Retailer A and (*b*) Retailer B. Normalized product unit price versus normalized product rating for the frequency of purchased products, whose rating is seen in the app. Frequency values are derived at the level of product categories. Note the increased frequency of purchased products with higher price as the product rating increases. This is particularly visible for Retailer A with a higher number of data records.

profile. Customers of Retailer B seem to have a more diverse demographic profile and a stronger interest in sustainability.

## 2.1. Relation between sustainability and price

During the field test, 16% of the field test consumers at both retailers pay at least 10 euro-cents more on average for each novel product by supporting at least one of the three environmental preferences (see electronic supplementary material, table S.1). This number is significant given that the price level of most products is low. Moreover, the percentage of consumers who regularly interact with the smart phone app (the ones in the centre of our causal impact analysis increases over time to 39% for Retailer A and 30% for Retailer B.

   We next assess whether the purchased products rated high within each product category are also the more expensive ones. First, both product deflated prices and the product ratings are normalized (see electronic supplementary material, § SN.2 for details). Figure 3 shows the frequency of the rated products that were viewed and purchased against price and rating. Retailer A clearly has a higher number of products with high prices and ratings as reflected by the high values on the top right of the heatmap. Similarly in Retailer B, for each normalized price value per product unit, a higher

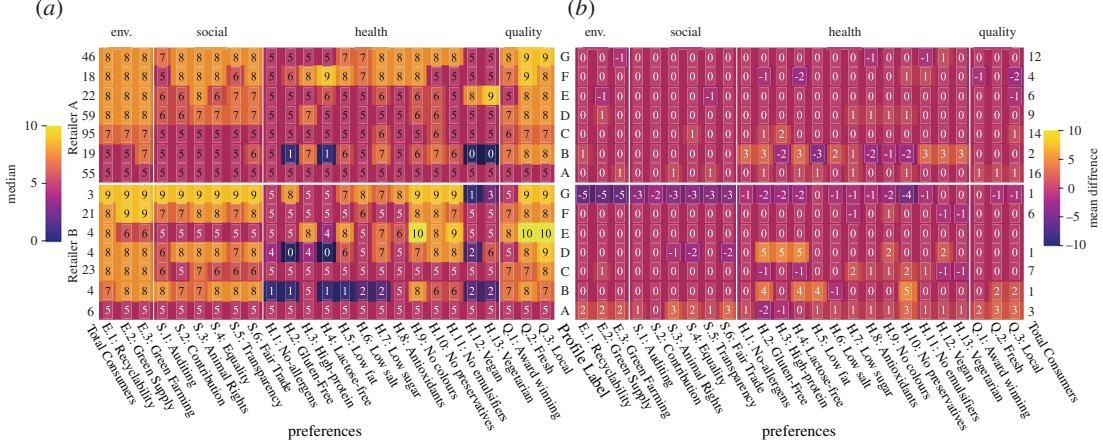

**Figure 4.** Preference score profiles over the four sustainability aspects for Retailer A (top) and Retailer B (bottom). Capital letters are labels relating to the extracted preference profiles. They are the same in both figures. The numbers on the left and on the right of each figure refer to the number of consumers in each profile. (a) Initial preference profiles demonstrating the high preference scores in environment and quality aspects. (b) Minimal targeted adjustments made during the field test confirming that product ratings match the expectations expressed via the sustainability preferences.

number of products are viewed and purchased that have higher rating values. We further measure the nonlinear relation between product price and product ratings with the Spearman's correlation coefficient across product categories. We observe a positive correlation coefficient of $\rho = 0.14$ and $\rho = 0.19$ for Retailer A ($p = 0.01$) and Retailer B ($p = 0.02$), respectively. This suggests that highly rated products may come with slightly higher prices.

The question that arises here is whether this positive correlation stems from the consumers' personalization, i.e. their choices regarding sustainability preferences, or the actual knowledge-base and ontology design that is universal for all consumers. To answer this question, we repeat these measurements, but instead of the normalized product ratings we used the max-min normalized sustainability index for each preference. The Spearman's correlation coefficient between the sustainability index of products and product prices per category is calculated for each preference across product categories. Out of 50 correlation values evaluated at each retailer, only one correlation value is greater than 0.5 and statistically significant. This confirms that no price biases are observed in the designed ontology in favour of more expensive products. Nevertheless, it is possible that certain preferences result in a sustainability index that is positively or negatively correlated with price due to the nature of this preference. For instance, vegetarian diet products may come with lower prices only because they do not contain meat that is usually expensive. See electronic supplementary material, tables S.12, S.13 and figure S.9 for an evaluation of all preferences.

## 2.2. Sustainability preference profiles

To understand how different consumers influence the product rating via personalization, we extract preference profiles by clustering consumers based on the preference choices they made during the field tests. We distinguish between *initial preferences* and *preference adjustments*. The initial preference choices are the very first ones made before consumers are exposed to the product rating functionality. They reflect the intrinsic intentions of consumers towards the sustainability aspects. The preference adjustments are made throughout the field test after the initial preferences were set.

*Initial preferences.* The profiles of the initial preferences are extracted by clustering consumers using the Ward hierarchical clustering approach [37]. Seven clusters are calculated for each retailer as shown in figure 4a. Note that most of the profiles consist of choices with a significantly high score in 'environment' and '(product) quality' as well as a high score in the category 'social'. The scores for 'health' shows a significant variation among the different profiles that reflects the diversity of dietary needs: (i) two neutral profiles with scores around 5 are observed for both retailers, (ii) there is a profile that penalizes strict preferences such as vegan, gluten/lactose-free, etc., with low score, and (iii) the majority of the profiles retain a high score for health indicators such as no preservatives/ colours/emulsifiers, etc. The majority of consumers avoids radical scores in health and expresses their sustainability profile via preferences regarding environment and quality aspects.

*Preference adjustments.* We assume that consumers adjust their preferences if they are not satisfied with how products are rated or if they aspire to explore new products. Consumers may also try to match the rated products with the ones they use to buy. We measure the mean difference between the initial preferences and all adjusted preferences per profile to understand how different consumer groups adjust their preferences to meet their expectations. Figure 4*b* shows the profile adjustments. We observe the following: (i) the majority of preference scores is on average unaffected, indicating that consumers usually do not need to adjust their preferences and they preserve their initial intrinsic sustainability preferences, which also validates the ontology from the consumers' perspective, (ii) consumers with initially neutral preferences increase their preference scores, especially in Retailer B, and (iii) consumers who initially maximize the preference scores slightly roll back to more moderate preference values, especially at Retailer B. Apparently, this rollback is related with a consumers' openness towards experiencing radically different highly rated products or with the ontology design and rebound effects that results in buying unexpected products. Similarly, the group of consumers that initially penalizes health preferences with low scores makes adjustments to higher scores.

# 3. Discussion and outlook

The findings of this work have several implications for consumers, retailers and producers. They also have an impact on science, policy and at an institutional level. We outline here the challenges and the new pathways that this work opens up.

*Consumers.* We show that a value-sensitive, preference-based design empowers more informed, transparent and accountable shopping choices tailored to personal sustainability values. As a result, consumers explore and finally buy new products. Compared to related work limited to health or environmental indicators for sustainability, we study a broader, yet finite set of such indicators (preferences). In the future, consumers themselves may expand or even limit this set according to their intrinsic priorities. Augmented shopping processes with smart phones may still disrupt regular shopping routines for some consumers. It is unclear to what extent and scale consumers can adopt such technologies at this moment. However, this was also the case with online shopping a few years ago and nowadays efforts such as Amazon Go [38] showcase the feasibility of such technologies in the market. The applicability of the value-sensitive shopping assistant in the context of online shopping is promising and may simplify its use by consumers. In terms of consumers' trust on products rating, new models are required to preserve a satisfactory level of explainability in the long run. This is especially the case when such models need to capture an expanding spectrum of product characteristics as well as the dependencies and incompatibility of sustainable development goals [10]. The transparency and accountability of the knowledge-base can be further enhanced with blockchain solutions as a means to securely verify sustainable production practices as well as the traceability of supply chains [39,40]. Recent studies also confirm how indispensable the active consumers' involvement is to evolve a viable and accurate knowledge-base on sustainability via citizen science initiatives [2,27].

*Retailers.* Retailers can pioneer future consumption patterns by offering products that are ethically more aligned to consumers' sustainability preferences. They can focus on selling a higher number of highly rated products that might come with slightly higher prices (or profit margins), while they serve consumers' values on sustainability.

*Producers.* Value-sensitive design provides new opportunities for producers to offer more attractive, higher quality products to retail markets, by capturing (i) consumers' sustainability preferences and (ii) reducing the sustainability gap of existing products. Moreover, producers have new opportunities to sell higher quality products at a higher price. This incentivizes the improvement of production practices, considering consumer preferences regarding environment, health or worker rights. In other words, thanks to welcomed recommendations of higher-quality products, more sustainable production can pay off.

*Science.* By demonstrating the shift of individuals to sustainable consumption, we also open up new opportunities for social coordination: moving from individual sustainability preferences to collective actions, which meets sustainable development goals in a bottom-up way. Scientific methods from the areas of game theory, mechanism design, incentive design, socio-technical optimization and learning [41–44] are becoming applicable to further support sustainability movements in society. Methods such as the ones of this work and a transdisciplinary scientific approach are required to improve and accelerate the development of sustainability knowledge-bases, using responsible AI and the wisdom of crowds [45,46].

*Policy-making and institutions.* We foresee new opportunities for environmental non-profit organizations to educate and interact with the general public by publishing their own personalization templates for

consumers, for instance, adopting the WWF or Greenpeace priorities for sustainability. Such customizable personalization is supported by the built system. Legislation could protect the open and public good character of a knowledge-base for sustainability. Such a knowledge-base should not be used as an unquestionable universal ground truth for sustainability. Instead, we envision the active, global involvement of various stakeholders such as public organizations, scientists and especially citizens to capture and develop the evolving nature of sustainability values. Promising related initiatives include the European Commission Environmental Footprint initiative or the Global Product Classification standardizing institution GS1. In other fields such as biology, initiatives for ontologies and knowledge-bases have had significant impact, for instance, the Gene Ontology Consortium [25]. Overall, the EU project ASSET [16], in the context of which this work has been carried out, is a promising blueprint showing how sustainable consumption might be scaled up in a participatory way.

# Methods

Here, we illustrate more details of the field tests and the causal impact analysis as well as the localization solution at retailer shops. We also outline the ontology design, the product rating methodology, the preference profiles and preference interactions.

## Field tests and causal impact analysis

*Recruitment.* Each shopping visit at retailer shops is rewarded with 10 euros. By analysing the history of purchases, 10 visits are expected on average during the field tests and therefore the total maximum amount of 100 euros incentivizes and rewards regular shopping behaviour. The study has received ethical approval by the ethical committee of ETH Zurich. To minimize biases originated by different human explanations of how to use the smart phone app (see electronic supplementary material, figure S.11), an integrated tutorial requires completion before consumers start using the smart phone app. A FAQ based on questions that arose during living laboratory experiments before the field tests was also included.

*Causal impact analysis.* The dynamic time warping (DTW) [35] algorithm is used to compare the time-series data of consumers that used the smart phone app (treatment consumer) against all consumers that did not use the smart phone app (non-treatment consumers) with purchase records up to 13 months before the field tests (excluding the last month as a buffer zone). Each comparison results in the similarity ranking of consumers, who did not use the smart phone app, from which the top-$k$ nearest neighbours are selected to form the group for assessing sustainable consumption. Since different values of $k$ yield different groups for comparison, the $k$ is selected via a DTW of weekly expenditure before the treatment as shown in electronic supplementary material, table S.9.

The temporal distance between consumers is measured over the three matching criteria (behaviour covariates). More specifically, the Euclidean distance with DTW is used to determine the top-$k$ nearest neighbours for the deflated monthly total budget spent and the sustainability index of the average purchased products per month. By contrast, the Jensen–Shannon distance (divergence) [47] with DTW is used for the distribution of deflated monthly budget spent per product category as such distance metric is known for measuring the similarity of distributions. For each consumer that used the smart phone app (treatment consumer), we rank all other consumers that did not use the app (non-treatment consumers) according to the three distance metrics. Since there are three distance metrics and we cannot determine which one has higher influence, the average ranking of the consumers that did not use the app is calculated over all combinations of distance metrics.

## Localization of products

A privacy-preserving indoor localization system is designed for the two retailer shops [22]. It derives the list of products that are in close proximity to consumers. It relies on Bluetooth low-energy beacons installed on the shop floor of both retailers. The beacons are aligned with the shop map to provide absolute anchors to the shop coordinates. By analysing the signal strength of the beacons using triangulation and a map logic on feasible positions and consumer movements, the position of the consumers in the shop is estimated, see figure 1a. This consumer position is related to the coordinates of product groups in close proximity. The rating of these products is calculated and presented to consumers' smart phone as shown in figure 1c. The localization system is privacy-preserving as all calculations are performed at the consumer's smart phone. Moreover, beacons do not obtain any information from consumers and therefore retailers cannot

derive consumer movements from this localization system. The deployed technology uses off-the-shelf hardware which yields low hardware costs. The obtained precision has an inaccuracy of about 2 m. The granularity of product group positions ranged from several metres down to zero accounting for the intrinsic ambiguity of product groups positions on top of each other.

## Ontology design

*Primitive concepts, tags and rules.* The ontology is designed to quantify the association between products and preferences, e.g. to what extent a certain product is for a vegetarian diet, fair trade etc. To measure such associations, we introduce a common alphabet of characteristics for products and preferences. This alphabet is a set of *keywords (tags)* that represent *primitive concepts*. They form the semantic space and scope of sustainability. Subsets of these primitive concepts compose a vocabulary of product and preference tags, while a primitive concept is not further decomposed to keep the ontology practical and consistent [48] within its scope, i.e. a primitive concept is regarded disjoint within a chosen scope of sustainability. For example, the preference tag 'vegan' can be composed by the two primitive concepts 'production with no animals' and 'production with no animal products'. Moreover, products and preference tags are assigned to products and preferences respectively based on logical rules. For instance, to assign the product tag 'low fat' to a product, a logical rule could determine the number of grams of fat, relative to the total product weight, contained in the product and/or whether the product has a low-fat label. Over 600 such rules are created for this purpose using the Drools framework [49]. We focused on food products. They are the ones that populate the knowledge-base and connect to product tags. In summary, the design of the ontology consists of (i) the choice of the primitive concepts, (ii) their assembly to product and preference tags, and (iii) the creation of logical rules to connect the tags with products and preferences. These actions required domain knowledge from experts (WWF, Greenpeace, Ethical Consumer, VKI), reliable data sources (e.g. EU reports) as well as the wisdom of the crowd by running the Social Impact Data Hack [50] to mine and structure information from Wikipedia, for instance, branding information (see electronic supplementary material, § SM.4).

*Association scores.* The association between a product and a preference tag is measured by their shared primitive concepts that satisfy a preference tag. We distinguish between positive and negative associations by determining for each pair of product and preference tags subsets of primitive concepts that semantically support or oppose the preference tag, i.e. the preference tag 'vegan' supports the primitive concept 'no animals involved in production' but opposes the concept 'animal product'. Therefore, the association score comes with positive and negative values in the range $[-1, 1]$ by summing up the associations between supported and opposed primitive concepts (see electronic supplementary material, § SM.2 for more details).

*Reduction design principle.* The construction of product and preference tags should adhere to the *reduction design principle*: (i) between two tags with the same primitive concepts, one and only one should be assigned to a product or preference, and (ii) when two tags assigned to a product or preference share primitive concepts, these primitive concepts should be removed and form a new tag. In the example of figure 5, the reduction design principle is violated if the product tag *AC* is assigned to the product, or, the preference tag *BC* is assigned to the preference. We prove in electronic supplementary material, § SM.1 how this principle minimizes the error of overlapping tags when the association scores are aggregated to calculate the rating of a product. Violations of the reduction design principle may result in excessive influence of certain preferences on the product rating. In practice, these artefacts may be captured by consumers, whose adjustments of preferences provide additional countermeasures against the error of semantically overlapping tags.

*Experts' guideline.* We propose a high-level guideline to populate the sustainability knowledge-base according to the proposed ontology. This guideline can be used by domain experts to guide the construction process and is outlined as follows:

1. Identify relevant primitive concepts based on (i) the product categories, (ii) the available product data, and (iii) the scope of sustainability preferences (goals).
2. Create product and preference tags using the primitive concepts such that these tags represent how product/preference characteristics oppose or support a product/preference.
3. Create rules that connect the product tags with products, and the preference tags with preferences.
4. Apply the reduction design principle between all combinations of product tags and preference tags that have overlapping concepts.

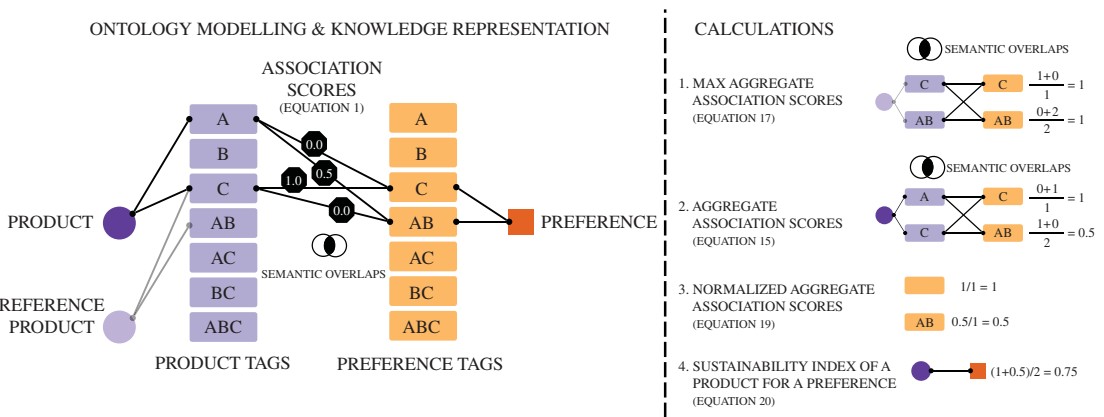

**Figure 5.** Calculation of the sustainability index using the ontology of products and preferences. Assume an alphabet of primitive sustainability concepts represented in this simple example by {A, B, C}. Combining the primitive concepts results is the word vocabulary {A, B, C, AB, AC, BC, ABC} of product and preference tags. Using rules based on experts' knowledge and verified data such as ingredients of products, the product tags A and C are assigned to a product. Similarly, a sustainability preference is designed by a composition of the two preference tags C and AB. We can now calculate the association scores between the product and preference tags in an automated way (without expert knowledge) by taking the intersection ∩ of the tags sets as follows: $|\{A\} \cap \{C\}|/|\{C\}| = 0$, $|\{C\} \cap \{C\}|/|\{C\}| = 1$, $|\{A\} \cap \{AB\}|/|\{AB\}| = 0.5$, $|\{C\} \cap \{AB\}|/|\{AB\}| = 0$. The sustainability index of a product for a given preference is calculated by the average normalized aggregate association scores of the assigned preference tags as demonstrated in this numerical example. The equation labels refer to the equations in electronic supplementary material, §SM.1.

5. Calculate association scores between product-preference tag pairs in the range [−1, 1].
6. If necessary, go back to Step 1, add or remove primitive concepts and repeat the process.

In Step 1, experts determine the scope of the sustainability by defining primitive concepts that capture key characteristics of products and preferences. Experts need to be aware of the products for which they design the ontology, the available information they have about these products as well as the sustainability preferences that should be captured. In Step 2, they can start combining these concepts into tags with the purpose of representing support or opposition to product characteristics and preferences. In Step 3, experts can assign these tags to product and preferences and can formalize rules under which these assignments are made. The processes of the first three steps are the most tedious ones and require knowledge, experience and a good overview of the available information. As an example of facilitating such processes, we performed workshops with several stakeholders during the project lifetime and organized the Social Impact Data Hack [50]. Step 4 applies the reduction design principle to improve the consistency of the ontology. Step 5 performs the calculations of the association scores based on the (automated) calculations illustrated in figure 5. In practice, these calculations are often calibrated by experts to reason about the association scores based on ground truth knowledge. For instance, consider a study that shows evidence about the effect of different preservatives on health. Obviously, the cause of such effects may be related to chemical or biological phenomena at a very low granularity level that is not captured within the scope of the designed sustainability ontology. In this case, association scores measuring can be calibrated to reflect the relative effect of preservatives according to the findings of such a study. Finally, the process can repeat by adding or removing primitive concepts. The motivation for this iterative process is to better capture the whole range of preferences, decompose further primitive concepts to make the ontology more granular, add/remove rules, expand product categories or enrich the knowledge-based with new datasets. During the ontology design, we performed over 10 iterations for validation purposes and the quality criterion for convergence was how well the product rating could be justified to consumers during the preliminary living laboratory tests.

## Product rating: sustainability index and preferences

*Sustainability index.* It quantifies the support or opposition of a preference by a set of product characteristics found in a product. This support or opposition is compared to a product, existing or hypothetical ('reference product' in figure 5), that has all possible characteristics that can support or

oppose respectively a preference. Figure 5 illustrates the involved calculations. The sustainability index between a product and a preference is measured using the normalized association scores aggregated over the connected product and preference tags. The normalized aggregated association score of a product-preference pair is the normalized aggregated association score of a product averaged over all preference tags assigned to the preference (Calculation 4 in figure 5). Each normalized aggregated association score between a product and a preference tag (Calculation 3 in figure 5) is calculated by the aggregated association scores of the product tags assigned to this product (Calculation 2 in figure 5) divided by the maximum association score between the reference product and the preference tags of the preference (Calculation 1 in figure 5).

*Insights on sustainability of production.* Note that by calculating for each preference the density of the sustainability index over all products, new opportunities arise to reason about the following: (i) the sustainability profile of different retailers, (ii) new ways (preferences) with which producers can improve products with a more sustainable profile, and (iii) market gaps where new business ecosystems can evolve with stronger involvement of producers and consumers to accelerate sustainable consumption. For instance, the densities in electronic supplementary material, figure S.10 confirm the more sustainable profile of Retailer B products across the preferences, e.g. higher sustainability index for animal rights, fair trade, recyclability and green farming. Improvements can be made by either introducing new products with better sustainability footprint over these preferences or by improving the existing production practices of the available products.

*Product rating.* Note that the sustainability index does not require any personal information for its calculation. It only relies on the information of the sustainability ontology, i.e. primitive concepts and tags, that we make available as public-good knowledge. As such, it can be calculated in public computational infrastructure, i.e. servers, public clouds, etc. By contrast, the calculation of the product rating requires personalization with consumers' preference choices that remain by design locally on their smart phones to protect privacy and limit manipulative nudging. As a result, the product rating is calculated on consumers' smart phones using the sustainability index values retrieved remotely using a distributed protocol of message passing between smart phones and a project server. The calculation is performed on-demand by consumers when they navigate in the retailer shop and request the rating of the products that are in their close proximity. For each product, the rating algorithm multiplies the sustainability index with the degree of opposition or support of each preference, measured by the distance (offset) from the median preference score (5, remaining neutral). These calculations are summed up and divided by the sum of all distances from the median preference scores. Electronic supplementary material, §SM.2 outlines in more detail the product rating calculation and its computational complexity. The (unscaled) product rating calculation is summarized as follows:

$$\text{product rating} = \frac{\text{sum of all (sustainability indices} * \text{preference offsets)}}{\text{sum of absolute preference offsets}}, \tag{4.1}$$

where the product rating values can be scaled to match different grading systems of different countries ([0, 10] in the field tests as supported in electronic supplementary material, § SM.2).

*Explainability.* Two levels of rating explainability are provided to consumers: (i) *product tags* and (ii) *preferences*. Consumers can learn about how each product characteristic influences the rating value by solving equation (34) of electronic supplementary material for a certain product tag, given that all other variables are known. Similarly, consumers can know how each offset of their preferences contributes to the product rating by solving equation (31) of the electronic supplementary material for a certain preference offset.

*Preferences selection.* The selection of preferences was made on the basis of providing a broad spectrum of different sustainability indicators with which consumers can express their preferences. However, this spectrum is not too broad to the extent of creating a cognitive overload for consumers and lack of comprehension about which preferences influence the rating of products and why. This is critical for the effectiveness of the rating explainability. Moreover, a lower number of preferences decreases the computational cost of the rating algorithm and improves the usability of the smart phone app (see electronic supplementary material, § SM.2.4). This balance is a result of the following process: (i) participation of several stakeholders in the ASSET project meetings and workshops providing insights about how grocery product choices influence different sustainability criteria, (ii) preliminary living laboratory experiments and smaller-scale field tests for feedback acquisition on the preferences, and (iii) choice based on available data, i.e. product and preference tags. Preferences with very similar preference tags or very few preference tags are removed or merged. The set of the final preferences shows a balance between several individual criteria on health (13) versus collective criteria in

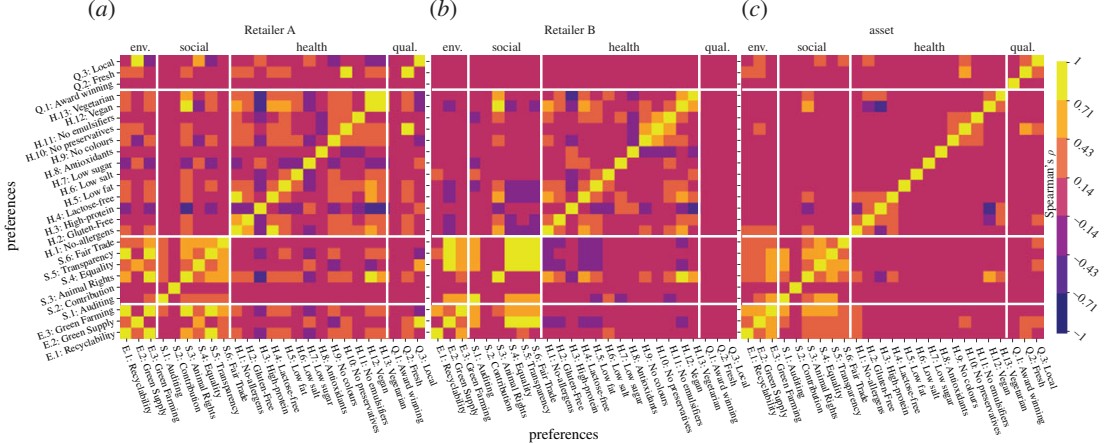

**Figure 6.** Interaction effects between different sustainability preferences. Ontology-level: correlation of association scores for (*a*) Retailer A and (*b*) Retailer B. Product-level: (*c*) correlation of sustainability indices for a common set of products between Retailer A and Retailer B. Positively correlated preference pairs across the *x* and *y*-axes indicate dependence whereas negatively correlated preference pairs indicate potential rebound effects.

environment, social and quality aspects (12). The consumers' feedback during the preliminary tests also determined the *strict preferences* of gluten-free, lactose-free, vegan and vegetarian products. Fully supporting such preferences results in excluding products that do not fully satisfy them even though they may satisfy other (non-strict) consumer preferences. In other words, strict preferences cancel association scores with other preferences.

## Preference profiles and interactions

*Preference profiles.* The Ward hierarchical clustering approach [37], with which the preference profiles are extracted, is selected based on its interpretable clusters and high scoring in the following evaluation metrics [51]: silhouette, Dunn index, Davies–Bouldin and Calinski–Harabasz. It uses Euclidean distance given that preference scores are continuous, not sparse, and different preference scores are comparable, with the exception of strict preferences that may introduce biases. No normalization or standardization is performed given the same scale among preferences.

*Interactions of preferences.* The sustainability preferences show correlations that originate from (i) the anticipated association between different product/preference tags (ontology-level) and (ii) the actual linking of products to product tags (product-level). This information at the ontology and product-level, respectively, can be used to reason about potential rebound effects [10]: undermining a sustainability preference by satisfying another one.

1. **Ontology-level**: The interaction of two preferences is measured as follows: (i) Find the shared product tags between the two preferences. (ii) For each of the two preferences and shared product tag, calculate the average association score among the preference tags of the preference and the shared product tag. (iii) Calculate the Spearman correlation coefficient of the average association scores among the two preferences.
2. **Product-level**: The interaction of two preferences at the product-level is measured as follows: (i) Calculate the sustainability index for each product-preference pair. (ii) Calculate the Spearman correlation coefficient of the sustainability indices among the two preferences.

These Spearman correlations are shown in figure 6. Potential rebound effects are observed between health/quality preferences versus environmental ones, while a highly rated product may be caused by multiple environmental preferences that share several common product tags.

Data accessibility. The ontology data are available here: https://doi.org/10.6084/m9.figshare.8313257. Data and relevant code for this research work are stored in GitHub: https://github.com/asikist/value-sensitive-design-code, and have been archived within the Zenodo repository: https://doi.org/10.5281/zenodo.4315081.
Authors' contributions. T.A. designed and implemented the ontology and product rating, carried out the data analysis, participated in the design of the study and drafted the manuscript; J.K. conceived and coordinated the project, participated in the design of the study and helped draft the manuscript; D.H. participated in the design of the

study, supervised the study and critically revised the manuscript; E.P. conceived the study, designed the study, supervised the analysis and drafted the manuscript.

Competing interests. We declare we have no competing interests.

Funding. This work was supported by the European Commission H2020 Program under the scheme 'ICT-10-2015 RIA' grant agreement #688364 ASSET–'Instant Gratification for Collective Awareness and Sustainable Consumerism' (https://www.asset-consumerism.eu) and by the LCM—K2 Center within the framework of the Austrian COMET-K2 program.

Acknowledgements. We thank the two retailers Coop and Winkler Markt for supporting the project and field tests as well as all consortium partners for their contributions (alphabetically): AINIA, BIA, Fastline and VKI. The authors are grateful to the participants of the Social Impact Data Hack [50] and in particular Maryna Pashchynska for her contribution to developing the sustainability ontology further. Several organizations supported this project in various ways such as WWF, Ethical Consumer, Greenpeace and Global2000. Finally, the authors would like to heartily thank in alphabetical order George Lekakos, Marloes Maathuis, Heinrich Nax, Michael Siegrist and Stefan Wehrli for their invaluable suggestions and feedback.

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

**14**