## [Reviewer comments · Royal Society Open Science]

Review History

RSOS-201418.R0 (Original submission)

Review form: Reviewer 1

Is the manuscript scientifically sound in its present form?

Yes

Are the interpretations and conclusions justified by the results?

Yes

Is the language acceptable?

Yes

Do you have any ethical concerns with this paper?

No

Have you any concerns about statistical analyses in this paper?

No

Recommendation?

Accept with minor revision (please list in comments)

Comments to the Author(s)

Overall, I enjoyed reading the manuscript, in particular, I like the paper idea and the motivation. The paper presents a relatively thorough methodology, data collection, and data evaluation.

Considering this, I think the authors did a good job of describing them.

Below I highlight several minor issues, which I had problems to fully understand:

Figure 2: It was not fully clear to me what exactly is the predicted value (e.g. what is the prediction horizon)? Why is the expenditure predicted? How exactly is this Figure illustrating the shift to sustainable consumption (it seem that weekly expenditure, at least with retailer A, was already growing before the field test)? What is the value of cumulative effect? It seems that the text is not commenting on cumulative effect (absolute affect if menioned on line 31). Weekly expenditure with retailer A and retailer B seem to be behave very differently. Any idea why?

pg. 6, lines 30-36. On those lines one can find text decribing the shift to sustainable behavior which is referring to various percentages. I am wondering whether values of those percentages are linked to values presented in Figures (e.g. Figure 2). I was not able to establish this link.

Figure 4: do the rows in 4a and 4b correspond to the same clusters of consumers? In Figure 4a they are marked with numbers, while in 4b capital letters are used. How large is each cluster?

Review form: Reviewer 2 (Nadine Sonnenberg)**Is the manuscript scientifically sound in its present form?**

Yes

Are the interpretations and conclusions justified by the results?

Yes

Is the language acceptable?

Yes

Do you have any ethical concerns with this paper?

No

Have you any concerns about statistical analyses in this paper?

Yes

Recommendation?

Accept with minor revision (please list in comments)

Comments to the Author(s)

I read through this manuscript with great interest and concluded that the study addresses novel research which will certainly appeal to a broad spectrum of readers - particularly those in support of a multi-disciplinary approach to deliver practical "real-world" solutions in the overarching pursuit of sustainable consumption. The focus of this study relates to the development of a smartphone app that can be used in supermarket setting to make informed choices in terms of sustainability criteria, via a platform that is trustworthy and respects the privacy of consumers.

Abstract:

The abstract is well written and immediately draws interest. I would suggest clarifying that the “novel personal shopping assistant” in fact refers to a smartphone app because this only becomes apparent after reading further through the introduction of the paper.

Introductory sections:

Similar to the comment relating to the abstract, I would suggest clarifying early on what the object of the design is because in reaching lines 52-53 (page 1) a reader may still be left perplexed as to what was designed in a value-sensitive manner. (Bear in mind that the title of the manuscript also does not offer an indication that the design actually relates to a smartphone app). From a social science/ consumer psychology and behaviour perspective, the “value” concept has many connections, which might stretch somewhat beyond the scope of what it was intended to describe in this study (e.g. values might describe a person’s enduring beliefs and other deeper socio-psychological motives that influence decision-making – lines 24-26 on page 2 is a typical example where “values” could very well be interpreted as such. On page 4, in Figure 1, a referral is made to “personal values” which seems even more in line with the socio-psychological understanding of values). In this regard, it might be worthwhile considering another term e.g. lines 16-17 (page 2) explains that the app was designed to rate products according to consumers’ preferences. “Preference” is perhaps a more accurate description than values and one could describe the app as “preference-based”, rather than “value-sensitive”.

Following through on the preferences highlighted on page 2 (lines 17-19), it would benefit the reader to offer a bit more explanation surrounding the chosen preference categories i.e. environment, health, social and quality. Quality, in particular, has a broad range of interpretations and it would be important to understand how the concept was incorporated into the consumer’s preference rating.

Results:

On page 5 (lines 43-53), results are reported that were derived from two different retail settings (A and B). It would be important to provide a brief description of the type of retail setting (e.g. discount supermarket versus speciality store) since consumers’ expectations and preferences may vary greatly depending on the type of store in question and also because retailers’ offerings tend to differ in terms of availability, price, assortment and many other facets – ultimately all of these impact on consumers’ purchase decisions. Although it would not be necessary to provide extensive detail on all of these facets, a brief description would suffice in highlighting the context in which data was collected.

The importance of price in consumers’ decisions to acquire products (and also more specifically sustainable products), should not be underestimated and deserves scrutiny. However, the results reported on page 5 that focus on the relation between price and sustainability is rather “unexpected” since no literature or background surrounding the importance of price was featured in the introductory sections of the manuscript. In this regard, I would recommend that a few sentences be added to the initial sections of the manuscript that develops a brief platform for the results reported in this section.

Minor comment: lines 44-47 on page 5 are rather difficult to follow and may benefit from being phrased somewhat differently to ensure greater clarity in the interpretation of the results. Also, the objective specified in lines 48-49 (page 5) seem to differ somewhat from those specified on page 4 (lines 32-38)?

Discussion and outlook:

The paper presents some interesting views on the possible applications of the technology that was developed and I believe these will be well received.

Decision letter (RSOS-201418.R0)

Dear Dr Klinglmayr

On behalf of the Editors, we are pleased to inform you that your Manuscript RSOS-201418 "How Value-Sensitive Design Can Empower Sustainable Consumption" has been accepted for publication in Royal Society Open Science subject to minor revision in accordance with the referees' reports. Please find the referees' comments along with any feedback from the Editors below my signature.

Please submit your revised manuscript and required files (see below) no later than 7 days from today's (ie 23-Nov-2020) date. Note: the ScholarOne system will 'lock' if submission of the revision is attempted 7 or more days after the deadline. If you do not think you will be able to meet this deadline please contact the editorial office immediately.

on behalf of Professor Matjaz Perc (Associate Editor) and Marta Kwiatkowska (Subject Editor)
openscience@royalsociety.org

Reviewer comments to Author:
Reviewer: 1

Comments to the Author(s)

Overall, I enjoyed reading the manuscript, in particular, I like the paper idea and the motivation. The paper presents a relatively thorough methodology, data collection, and data evaluation. Considering this, I think the authors did a good job of describing them. Below I highlight several minor issues, which I had problems to fully understand:

Figure 2: It was not fully clear to me what exactly is the predicted value (e.g. what is the prediction horizon)? Why is the expenditure predicted? How exactly is this Figure illustrating the shift to sustainable consumption (it seem that weekly expenditure, at least with retailer A, was already growing before the field test)? What is the value of cumulative effect? It seems that the text is not commenting on cumulative effect (absolute affect if menioned on line 31). Weekly expenditure with retailer A and retailer B seem to be behave very differently. Any idea why?

pg. 6, lines 30-36. On those lines one can find text decribing the shift to sustainable behavior which is refering to various percentages. I am wondering whether values of those percentages are linked to values presented in Figures (e.g. Figure 2). I was not able to establish this link.

Figure 4: do the rows in 4a and 4b correspond to the same clusters of consumers? In Figure 4a they are marked with numbers, while in 4b capital letters are used. How large is each cluster?

Reviewer: 2

Comments to the Author(s)

I read through this manuscript with great interest and concluded that the study addresses novel research which will certainly appeal to a broad spectrum of readers - particularly those in support of a multi-disciplinary approach to deliver practical "real-world" solutions in the overarching pursuit of sustainable consumption. The focus of this study relates to the development of a smartphone app that can be used in supermarket setting to make informed choices in terms of sustainability criteria, via a platform that is trustworthy and respects the privacy of consumers.

Abstract:

The abstract is well written and immediately draws interest. I would suggest clarifying that the "novel personal shopping assistant" in fact refers to a smartphone app because this only becomes apparent after reading further through the introduction of the paper.

Introductory sections:

Similar to the comment relating to the abstract, I would suggest clarifying early on what the object of the design is because in reaching lines 52-53 (page 1) a reader may still be left perplexed as to what was designed in a value-sensitive manner. (Bear in mind that the title of the manuscript also does not offer an indication that the design actually relates to a smartphone app). From a social science/ consumer psychology and behaviour perspective, the "value" concept has many connections, which might stretch somewhat beyond the scope of what it was intended to describe in this study (e.g. values might describe a person's enduring beliefs and other deeper socio-psychological motives that influence decision-making - lines 24-26 on page 2 is a typical example where "values" could very well be interpreted as such. On page 4, in Figure 1, a referral is made to "personal values" which seems even more in line with the socio-psychological understanding of values). In this regard, it might be worthwhile considering another term e.g. lines 16-17 (page 2) explains that the app was designed to rate products according to consumers' preferences. "Preference" is perhaps a more accurate description than values and one could describe the app as "preference-based", rather than "value-sensitive".

Following through on the preferences highlighted on page 2 (lines 17-19), it would benefit the reader to offer a bit more explanation surrounding the chosen preference categories i.e. environment, health, social and quality. Quality, in particular, has a broad range of interpretations and it would be important to understand how the concept was incorporated into the consumer's preference rating.

Results:

On page 5 (lines 43-53), results are reported that were derived from two different retail settings (A and B). It would be important to provide a brief description of the type of retail setting (e.g. discount supermarket versus speciality store) since consumers' expectations and preferences may vary greatly depending on the type of store in question and also because retailers' offerings tend

to differ in terms of availability, price, assortment and many other facets – ultimately all of these impact on consumers' purchase decisions. Although it would not be necessary to provide extensive detail on all of these facets, a brief description would suffice in highlighting the context in which data was collected.

The importance of price in consumers' decisions to acquire products (and also more specifically sustainable products), should not be underestimated and deserves scrutiny. However, the results reported on page 5 that focus on the relation between price and sustainability is rather "unexpected" since no literature or background surrounding the importance of price was featured in the introductory sections of the manuscript. In this regard, I would recommend that a few sentences be added to the initial sections of the manuscript that develops a brief platform for the results reported in this section.

Minor comment: lines 44-47 on page 5 are rather difficult to follow and may benefit from being phrased somewhat differently to ensure greater clarity in the interpretation of the results. Also, the objective specified in lines 48-49 (page 5) seem to differ somewhat from those specified on page 4 (lines 32-38)?

Discussion and outlook:

The paper presents some interesting views on the possible applications of the technology that was developed and I believe these will be well received.

===PREPARING YOUR MANUSCRIPT===

===PREPARING YOUR REVISION IN SCHOLARONE===

To revise your manuscript, log into <https://mc.manuscriptcentral.com/rsos> and enter your Author Centre - this may be accessed by clicking on "Author" in the dark toolbar at the top of the

page (just below the journal name). You will find your manuscript listed under "Manuscripts with Decisions". Under "Actions", click on "Create a Revision".

<https://royalsociety.org/journals/authors/author-guidelines/#supplementary-material> to include a suitable title and informative caption. An example of appropriate titling and captioning may be found at https://figshare.com/articles/Table_S2_from_Is_there_a_trade-off_between_peak_performance_and_performance_breadth_across_temperatures_for_aerobic_sc_ope_in_teleost_fishes_/3843624.

Author's Response to Decision Letter for (RSOS-201418.R0)

See Appendix A.

Decision letter (RSOS-201418.R1)

Dear Dr Klinglmayr,

It is a pleasure to accept your manuscript entitled "How Value-Sensitive Design Can Empower Sustainable Consumption" in its current form for publication in Royal Society Open Science. The comments of the reviewer(s) who reviewed your manuscript are included at the foot of this letter.

on behalf of Professor Matjaz Perc (Associate Editor) and Marta Kwiatkowska (Subject Editor)
openscience@royalsociety.org

Associate Editor Comments to Author (Professor Matjaz Perc):

Thank you for the comprehensive revision of your manuscript, which we are happy to accept for publication in Royal Society Open Science.

Appendix A

How Value-Sensitive Design Can Empower Sustainable Consumption

Responses to reviewers

We would like to thank all reviewers for their effort to review our manuscript thoroughly as well as for their invaluable feedback. Below are our responses to each reviewer's comment. The changes in the revised manuscript version have been marked with red color.

Reviewer: 1

“Figure 2: It was not fully clear to me what exactly is the predicted value (e.g. what is the prediction horizon)? Why is the expenditure predicted? How exactly is this Figure illustrating the shift to sustainable consumption (it seem that weekly expenditure, at least with retailer A, was already growing before the field test)? ”

The weekly expenditure for sustainable products, meaning the ones rated higher than 5, is the criterion to demonstrate the shift to more sustainable consumption. The actual weekly expenditure value is therefore compared to the predicted one during the field test. Therein, the weekly expenditure is predicted based on historic data to estimate the shopping behavior of the treatment group in case they did not use the shopping app. Actual vs. predicted values for both retailer shops confirm a statistically significant shift of expenditures to highly rated products. Clarifications about all these aspects are made in the text illustrating Fig. 2.

“What is the value of cumulative effect? It seems that the text is not commenting on cumulative effect (absolute affect if mentioned on line 31). ”

We now clarify the role of the cumulative effect reflecting how the shift of expenditures towards more sustainable products unfolds over the period of the field test.

“Weekly expenditure with retailer A and retailer B seem to be behave very differently. Any idea why?”

We have added some interesting information about the profile of the two retailers that are associated with the observed patterns.

“pg. 6, lines 30-36. On those lines one can find text describing the shift to sustainable behavior which is referring to various percentages. I am wondering whether values of those percentages are linked to values presented in Figures (e.g. Figure 2). I was not able to establish this link.”

The percentage values provide additional information that is relevant for the interpretation and validation of the causal effects presented in Fig. 2: the products used in the analysis were actually seen in the app before purchase, indicating that the users did use the app before buying the products. We have split the text in paragraphs to make the distinction clearer and we also explain the role of the percentage values now.

“Figure 4: do the rows in 4a and 4b correspond to the same clusters of consumers? In Figure 4a they are marked with numbers, while in 4b capital letters are used. How large is each cluster?”

We now clarify in the figure caption that the capital letters are labels for the extracted preference profiles and they are the same in the two figures. The numbers on the left and on the right of each figure refer to the number of consumers in each profile.

Reviewer: 2

“Abstract: The abstract is well written and immediately draws interest. I would suggest clarifying that the “novel personal shopping assistant” in fact refers to a smartphone app because this only becomes apparent after reading further through the introduction of the paper.”

We have added that the novel personal shopping assistant is implemented as a smart phone app.

“Similar to the comment relating to the abstract, I would suggest clarifying early on what the object of the design is because in reaching lines 52-53 (page 1) a reader may still be left perplexed as to what was designed in a value-sensitive manner. (Bear in mind that the title of the manuscript also does not offer an indication that the design actually relates to a smartphone app)”

The smart phone app is now referred to earlier in the text of the Introduction. We have also made clarifications regarding the value-sensitive approach.

“From a social science/ consumer psychology and behaviour perspective, the “value” concept has many connections, which might stretch somewhat beyond the scope of what it was intended to describe in this study (e.g. values might describe a person’s enduring beliefs and other deeper socio-psychological motives that influence decision-making – lines 24-26 on page 2 is a typical example where “values” could very well be interpreted as such. On page 4, in Figure 1, a referral is made to “personal values” which seems even more in line with the socio-psychological understanding of values). In this regard, it might be worthwhile considering another term e.g. lines 16-17 (page 2) explains that the app was designed to rate products according to consumers’ preferences. “Preference” is perhaps a more accurate description than values and one could describe the app as “preference-based”, rather than “value-sensitive”. ”

Thank you for your excellent comment. We now use the term “preference” in relation to consumer priorities and choices over different sustainability criteria and the term “value” in connection with how the system is designed to address more universal societal goals such as privacy, sustainability, or self-determination. We also present our approach as value-sensitive and preference-based, now. “Preference-based” addresses the choices of consumers based on the value of self-determination.

“Following through on the preferences highlighted on page 2 (lines 17-19), it would benefit the reader to offer a bit more explanation surrounding the chosen preference categories i.e. environment, health, social and quality. Quality, in particular, has a broad range of interpretations and it would be important to understand how the concept was incorporated into the consumer’s preference rating.”

With “quality” we mean “product quality”. In our study, each preference category is measured in a particular way, with specific features. The related information about the different preference categories, what they mean and how they are determined is explained in the tables of the Supplementary Information. We now explicitly refer to these tables in our main manuscript.

“On page 5 (lines 43-53), results are reported that were derived from two different retail settings (A and B). It would be important to provide a brief description of the type of retail setting (e.g. discount supermarket versus speciality store) since consumers’ expectations and preferences may vary greatly depending on the type of store in question and also because retailers’ offerings tend to differ in terms of availability, price, assortment and many other facets – ultimately all of these impact on consumers’ purchase decisions. Although it would not be necessary to provide extensive detail on all of these facets, a brief description would suffice in highlighting the context in which data was collected.”

We have added a paragraph related to Fig. 2, which gives some background information about the retailers and the seasonality effect. Further information is available in the Supplementary Information.

“The importance of price in consumers’ decisions to acquire products (and also more specifically sustainable products), should not be underestimated and deserves scrutiny. However, the results reported on page 5 that focus on the relation between price and sustainability is rather “unexpected” since no literature or background surrounding the importance of price was featured in the introductory sections of the manuscript. In this regard, I would recommend that a few sentences be added to the initial sections of the manuscript that develops a brief platform for the results reported in this section.”

The relation of sustainability with quality and price is now discussed in the introduction and backed up with a reference. Figure S.5-S.7 in the Supplementary Information illustrate survey answers regarding consumer beliefs and characteristics that can support the further interpretation of the results.

“lines 44-47 on page 5 are rather difficult to follow and may benefit from being phrased somewhat differently to ensure greater clarity in the interpretation of the results. ”

We have made some edits in this paragraph to improve its readability.

“Also, the objective specified in lines 48-49 (page 5) seem to differ somewhat from those specified on page 4 (lines 32-38)?”

We introduce some important information about price before we tackle the objective in the next paragraph starting as follows: The question that arises here is whether this positive correlation stems from the consumers’ personalization, i.e. their choices regarding sustainability preferences, or from the actual knowledge-base and ontology design that is universal for all consumers.